# Pediatric Renal Cell Carcinoma (pRCC) Subpopulation Environmental Differentials in Survival Disadvantage of Black/African American Children in the United States: Large-Cohort Evidence

**DOI:** 10.3390/cancers16233975

**Published:** 2024-11-27

**Authors:** Laurens Holmes, Phatismo Masire, Arieanna Eaton, Robert Mason, Mackenzie Holmes, Justin William, Maura Poleon, Michael Enwere

**Affiliations:** 1Public Health & Allied Health Science Department, Wesley College, Delaware State University, Dover, DE 19901, USA; rmason@desu.edu; 2Biological Sciences Department, University of Delaware, Newark, DE 19716, USA; 3Medical College of Wisconsin, Milwaukee, WI 53226, USA; 4College of Population Health, Thomas Jefferson University, Philadelphia, PA 19144, USA; phatsimo.masire@jefferson.edu; 5Global Health Equity Foundation, Bear, DE 19701, USA; holmesmackenzie04@gmail.com (M.H.); maurapoleon@gmail.com (M.P.); mikky89@gmail.com (M.E.); 6Department of Integrated Physiology and Health Sciences, Alma College, Alma, MI 48801, USA; 7Texas A & M University, College Station, TX 77843, USA; 8PrimeLife360, Wellness 20 Wenlock Road, London N1 7GU, UK

**Keywords:** pediatric renal cell carcinoma, urbanity, household median income, black/AA survival, environmental differentials

## Abstract

Cancer remains the leading cause of mortality among children, 0–14 years in the United States, indicative of specific malignant neoplasm such as brain/CNS (glioblastoma, ependymoma, astrocytoma, etc.), chronic myeloid leukemia, reflecting late stage at diagnosis and therapeutic differentials, depending on aberrant epigenomic modulations. The pediatric renal cell carcinoma (pRCC) remains challenging during early childhood (0–4 years), indicative of the late stage at diagnosis and survival disadvantage. The current study utilized large cohort of children with RCC, USA National Cancer Institute SEER data and the application of retrospective cohort design and Cox Proportional Hazard modelling in survival disadvantage of subpopulations with pRCC. While survival advantage marginalized among blacks/AA children, relative to their White counterparts, household median income as well as urbanity (geographic locale) mainly urban area substantially marginalized survival outcome. These findings are suggestive of the availability of reliable pRCC diagnostics and therapeutics in rural and urban areas in the USA.

## 1. Introduction

Cancer remains the leading cause of death among children, 0–14 years old, in the US, with an estimated 1790 children and adolescents expiring in 2017 [1]. The incidence of renal cell carcinoma (RCC) has risen steadily in recent decades, with an estimated 5.4% each year [2,3]. In the pediatric population, RCC is the second most common form of renal malignancy, while Wilms Tumor remains the most common pediatric RCC (pRCC) [4]. Although RCC accounts for 2–12.5% of pediatric renal cancer diagnoses, previous studies estimated an overall age-adjusted incidence of 0.01/100,000 [2,3]. However, numerous studies have observed higher incidence among black/AA populations [3,5] as well as specific studies at individual patient and subgroup levels [6,7,8,9,10,11]. Silberstein et al. observed the pRCC period prevalence as cumulative incidence as three times as likely among black/AA populations with respect to the overall pediatric population in the US [4].

pRCC is most likely to be diagnosed in the second decade of life, with the odds of being diagnosed with this malignancy increasing by 50% annually after age 12 [5]. The RCC is indicative of over 70% of renal tumors in adolescence (>14 years) [5]. In the adult population, racial disparities exist in survival outcome, while five-year survival is estimated at 72.6% for white patients, compared with their black/AA counterparts, 68.0% [12,13].

Among adult patients, RCC risk determinants such as genetics, cigarette smoking, excess alcohol consumption, obesity, hypertension, and/or related medications have been implicated in RCC incidence. However, the pRCC risk determinants are not very fully understood [1]. Due to the range of clinical manifestations of pRCC, as well as numerous non-specific symptoms, pRCC may be misdiagnosed in early stages [2]. Rialon et al. observed higher survival among pediatric patients, diagnosed at early stages, indicative of 100% five-year survival at stage I, relative to 8% survival at stage IV [14]. However, this survival advantage could be explained by accessibility to radical nephrectomy and other types of surgery, which has a 95% five-year survival, compared to the 20% five-year survival of patients who did not undergo surgical resection [14].

Additional challenges arise from the differences in histopathology between pRCC and adult RCC patients, which had been observed in previous findings by Geller, Argani, et al. Translocational RCC is predominant among pRCC, accounting for approximately 33–72% of RCC diagnoses [13,15,16]. However, translocational RCC is resistant to traditional immunotherapy in RCC patient therapeutics, with a substantial prognostic marginalization of adult RCC patients [16].

With respect to healthography as urbanity, limited studies have been conducted to examine rural–urban disparities in cancer survival, such as timely diagnoses and early treatment [17,18,19]. Specifically, rural–urban disparities have also been observed in RCC among rural Illinois males, which is twice as likely compared with the U.S. population. These findings observed RCC stages as more regional and distant stages in rural Illinois males compared to the rest of the state [19,20]. At a national level, studies observed academic and urban hospitals with higher utilizations of partial nephrectomy for RCC than rural hospitals, which may lead to rural–urban disparities in RCC mortality [20,21].

Despite the availability of previous clinical and epidemiologic data with higher incidences of pediatric cancer among white children and survival disadvantages among black children, this correlation may not exist in pRCC incidence and survival at a national level. For example, Rialon et al. demonstrated no differences in gender or ethnicity/race between those who underwent surgical therapy and those who did not [14]. Additionally, much of the existing literature on pRCC are case reviews of 10–40 cases limited to one region or institution, reducing the applicability of any correlations between RCC and basic variables, including race or sex, to the national population level [17].

The current investigation aimed to address pediatric renal cell carcinoma (pRCC) nationally, as a representative sample of pRCC cumulative incidence, annual percent change, and survival, namely the Surveillance Epidemiology and End Results (SEER) registry, 1973–2015. This epidemiologic design utilized a retrospective cohort study in the assessment of subpopulation mortality and survival differentials. Specifically, population-based registries (SEER) on cancer incidence and survival within 18 states in the United States were utilized. The objective of this study was to investigate subpopulation variations in pRCC by pRCC demographic and clinical features, as well as the assessment of the temporal trends such as age-adjusted incidence, rate ratio, percent change, annual percent change, mortality, event-free survival (EFS), and survival differences by race, household annual median income, and area of residence as urbanity.

## 2. Materials and Methods

The Data Use Approval (DUA) was obtained from the National Cancer Institute (NCI), SEER registry, for data acquisition and the application of these data in the assessment of pediatric renal cell carcinoma subpopulations in survival.

### 2.1. Study Design

This study utilized a population-based registry, SEER, and the application of a non-experimental epidemiologic design, as a retrospective cohort study to assess pediatric RCC (pRCC) survival, as well as racial heterogeneity. Additionally, the incidence proportion, cumulative incidence, and trends were assessed using age-adjusted parameters. This epidemiologic design as a cohort study is reliable and accurate, given pre-existing data from the National Cancer Institute (NCI) Surveillance Epidemiology End Results (SEER) registry.

### 2.2. Data Source

We utilized data from NCI’s SEER registry in this study. The SEER program is a cancer registry operated and managed by the NCI. This registry began in 1973 with nine SEER areas. In 1992, SEER expanded to include four additional areas, while in 2005, SEER further expanded and included five additional areas, rendering the current registry in SEER as 18 areas. The information collected and stored in this registry includes cancer diagnosis, patient demographics, primary tumor site, tumor morphology and stage at diagnosis, prognostic factors, and vital status (dead or alive), as well as some social determinants of health (SDHs), namely education, household median income level, and area of residence. This registry remains the most comprehensive source of population-based cancer data, which includes cancer stage at the time of diagnosis as well as patient survival. The selection of the areas into the SEER program is based on the ability of cancer centers to provide high-quality, population-based, and other variables to the SEER registry.

In the SEER registry, cancers are coded according to the International Classification of Disease (ICD-O-3) 3rd edition. The SEER registries update these databases continuously and we used SEER Stat software to adjust for the delay in response rates. In this study, we presented incidence trends and survival estimates for three racial groups (white, black, and other). SEER cases are reported to NCI annually each November and there is a completion rate of approximately 98% for all site-specific malignancy, except melanoma.

The trend data were divided into two registry periods, namely 1973–2015 and 2000–2015. These categories were based on the SEER registries. The 1973–2015 data were the initial registries, which covered 9 SEER areas, while 2000–2015 embraced recent registries and covered 18 SEER areas. The survival data covered 1973–2015, implying the entire SEER registry; however, these data present different risks of dying based on the year of diagnosis.

### 2.3. Variable Ascertainment

The response, dependent, and outcome variable was pRCC survival (vital status), which was measured with survival months as the function of mortality. This survival time is indicative of the impact or force of dying, given time as an exposure function of pRCC survival. As a response variable, the hazard of dying remains constant over time from the onset of observation until the end of observation, where some in the cohort did not experience the event (lack of dying as survival) and some were censored.

The independent and confounding variables assessed were race, ethnicity, sex, age at diagnosis, area of residence, income, and education. Race is an exposure function of dying, given time as a contributing factor of dying. The race variable was recoded as white, black/African American (AA), and other, where other included American Indian/Alaskan Native, Asian/Pacific Islander, and unknown. Sex was recoded as male and female. Meanwhile, the age at diagnosis was captured as age groups, namely (a) infants (<1 year); (b) 1–4 years; 5–9 years; (c) 10–14 years; and (d) 15–19. This variable was addressed as a categorical scale in the model.

The area of residence as urbanity was recoded as urban, metropolitan, and rural. Due to residence being an explanatory variable in this study, the area of residence was recoded as metropolitan and urban, implying the integration of rural into the urban cluster. Median household income was recoded as families living below the poverty level. Using this variable, we recoded income into three categories, implying the (a) lowest, (b) moderate, and (c) highest income levels. Education was recoded as (a) low, (b) moderate, and (c) high education, implying the first level or category as the highest and the last level as the lowest education status. Income and education variables in this study were based on SEER-derived data from the census, which rendered these variables as aggregate or group data rather than individual data.

### 2.4. Sample Size and Power Estimation

The current study utilized pre-existing data on cases of children with RCC, *n* = 174, to estimate the power, implying the ability of pRCC to detect a clinically and biologically meaningful difference in the cumulative incidence, mortality, and survival, as well as causal inference (random error quantification—*p* value). The power of this study was estimated using the following parameters: (a) sample size (*n* = 174), categorized by racial subgroups, white (*n* = 116), black/AA (*n* = 49), and other races (*n* = 6); (b) effect size = 0.20 (20%) as the hazard ratio; and (c) type I error tolerance. And the precision measure was 95% CI, 0.05 for the univariable model as well as 99% CI, 0.01 for the multivariable model. With these parameters, we estimated the power at 99% (type I error tolerance < 1%), which is a sufficient power to detect a minimum difference of 10% comparing the mortality and survival experience of black/AA children relative to white counterparts with pRCC.

### 2.5. Cumulative Incidence Rate and Trend Analysis

A weighted least squares (WLS) method was used to estimate the age-adjusted RCC trends in age groups of children from <1 to 19 years. This method indicates assessment of the random errors in the model and is applicable to regression functions that are either linear or nonlinear in the parameter estimates of the sample statistic. Its application in model fitting incorporates extra non-negative constants, or weights, associated with each data point, into the fitting criterion. The size of the weight illustrates the precision of the information contained in the associated observation, implying the parameter value (point estimate) and parameter precision (*p* value). Therefore, optimizing the weighted fitting criterion to identify or examine the parameter estimates allows the weights to determine the contribution of each observation to the final parameter estimates. The percent change (PC) and annual percent change (APC) were estimated using SEER Stat Software. Percent changes were calculated using 1 year for each end point and APCs were calculated using weighted least squares methods.

The mortality estimates were analyzed using the Binomial Regression model in risk ratio parameters, as point estimates. The survival analysis utilized the Cox proportional hazard model, with the assumption that the hazard ratio remains constant over time. Prior to this analysis, pre-hypothesis test screening was performed to identify outliers and missing variables in the data. To summarize the categorical variables, frequency and percentages were utilized in this study.

To examine the survival proportion, Kaplan–Meier survival curves were utilized, while a Nelson–Aalen Cumulative Hazard was utilized in assessing the hazard of dying. To assess the quality of survival, the Log Rank test was used, which is a chi-square stratification analysis with a degree of freedom. In addition, the life table was applied in examining the five-year relative survival, comparing race and sex differences in pediatric RCC survival.

Prior to the survival analysis, Cox proportional hazard model assumption using the Global Test Schoenfeld (GTS) was applied. With STATA statistical software, this GTS model utilized this syntax: stphtest, detail. No application of this test is indicative of the Cox model assumption violation, which claims that the hazard ratio remains constant over time.

In examining the factors associated with survival, as well as the predictors of survival, the Cox proportional hazard model was utilized in a univariable model. Since prognostic factors such as tumor stage, pediatric age, etc., influence survival, a multivariable model was built after examining these variables for the potential confounding effect, as well as effect measure modification (EMM). In assessing the confounding variables, we utilized Mantel–Hansel as well as Cox–Mantel stratification analyses, which allowed for the examination of the crude and adjusted estimate variables, termed as confounding. The effect measure modifiers, namely area of residence and median household income, were assessed using a Cochran–Mantel–Hansel (CHM) stratification analysis, which allowed for the detection of the odds specific hazard and hazard ratio, respectively. To examine non-confounding, in the relationship between race and pRCC survival, a multivariable model was used after assessing area of residence as EMM. In this model, sex and age were entered as biologic and clinical variables, while predictors of survival were median household income and education, as SDHs. The multivariable models were built following these trajectories: Model I (race); Model II (race, age, sex); Model III (race, age, sex, insurance); and Model IV (race, age, sex, insurance, income, education). This final model was based on area of residence as EMM as a heterogeneity effect.

The type I error tolerance was set at 5% (0.05, 95% CI) for the univariable model, while the multivariable model was set at 1% (0.01, 95% CI). All tests were two-tailed. The trend and rate analyses were performed using SEER Stats version 8.3.5 (NCI, SEER), while the Binomial Regression model and survival analysis were performed with the Cox proportional model, using STATA statistical software, version 17.0 (STATA Corporation, College Station, TX, USA).

## 3. Results

These data described pediatric RCC (pRCC) using a representative sample from the National Institute of Health (NIH), National Cancer Institute (NCI), Survival Epidemiology End Result (SEER), between 1973 and 2015. Although not illustrated in the table, of the total sample of children diagnosed with RCC during this period (*n* = 174), 49 expired regarding mortality (28.2%). The overall sample comprised white (66.7%), black/AA (28.2%), and other (5.2%) races. With respect to sex, females were diagnosed more during this period, relative to males, 52.9% versus 47.1%, respectively. The majority of children diagnosed with RCC were in the age group of 15–19 (55.2%). The histologic characterization in terms of tumor grade indicated a higher proportion of unspecified and unknown pRCC (61.5%). In addition, tumor grade 2 was the second most prevalent cumulative incidence (16.7%). Regarding education, group level and aggregate data were recoded into three levels, and the lowest education level included 50.0%. Similarly, the income level represented group data recoded into tertial groups, with the highest tertial group representing 48.9%. The private medical insurance coverage represented 24.1% of the overall sample of children with RCC. The geographic locale as urbanity, implying the area of residence of pediatric patients with RCC, was classified as urban versus metropolitan, with the majority of cases within the metropolitan area, *n* = 159 (91.4%).

Table 1 demonstrates the study characteristics stratified by geographic locale (urbanity) as the area of residence of children during the time of the tumor diagnosis. A greater proportion of pRCC mortality occurred in the urban area, relative to the metropolitan area. With respect to race, white children in the urban area were more likely to experience greater proportional RCC morbidity relative to white children in the metropolitan area, 73.3% versus 66.0%, respectively. In contrast, black/AA children in the metropolitan area, relative to the urban area, were more likely to experience RCC, 28.9% versus 20.0%, respectively. There were more males diagnosed with pRCC in the metropolitan area compared to the urban area (52.2% versus 40.0%). In contrast, there were more females diagnosed with RCC in the urban area relative to the metropolitan area (60.0% versus 47.8%). Within the age group of 10–19, more pRCC was diagnosed in the urban area relative to the metropolitan area. The proportions of 3rd and 4th grade tumors diagnosed were greater in the urban area relative to the metropolitan area, indicative of a survival disadvantage in the urban area. The lowest-level-of-education proportion was higher in the urban area relative to the metropolitan area (66.7% versus 48.4%). The insurance did not indicate differences comparing urban to metropolitan areas. The lowest-income-level proportion was higher in the urban area, relative to the metropolitan area (33.3% versus 24.5%).

Table 2 illustrates the prognostic and sociodemographic factors as an exposure function of pRCC survival. Despite a high cumulative incidence, but as marginally precise as random error quantification, with a *p* value, children diagnosed with RCC (pRCC) in metropolitan areas were 53% less likely to die, hazard ratio (HR) = 0.47; 95% CI, 0.21–1.05; *p* = 0.065. There was racial variance in pRCC survival. Compared with white children, black/AA children with RCC were almost three times as likely to expire (die), HR = 2.90; 95% CI, 1.58–5.31. There was sex variability in pRCC survival. Relative to females, males were 21% more likely to die, HR = 1.21, 95% CI = 0.69–2.11. Age variance was observed in pRCC survival, despite statistical instability of the parameter precision value. Compared to children aged 15–19, children aged 1–4, 5–9, and 10–14 were 72%, 50%, and 21% less likely to expire. The tumor grade was associated with survival differentials; compared to children with a very poor tumor grade (pRCC grade 4), children diagnosed with grades 1, 2, and 3 were less likely to expire. Specifically, compared with grade 4, children with grade 1 were 62% less likely to die, while children with grade 3 were 31% less likely to die. Additionally, there was an inverse correlation between education level and risk of dying, since the lower the education level, the more likely the survival disadvantage, and hence excess pRCC mortality. Figure 1 illustrates the overall survival of children with RCC.

Figure 2 demonstrates the survival experience of children with RCCs by race/ethnicity. The lower the survival curve (line), the more the survival disadvantage, while the higher the survival curve, the better the survival advantage. Specifically, black/AA children with RCCs experienced survival disadvantage relative to their white counterparts.

Table 3 illustrates a non-confounding and adjusted association between area of residence and race as exposure functions of pRCC survival. Model I is indicative of an unadjusted association between race and pRCC survival, stratified by area of residence as urbanity. In this unadjusted model, relative to white children, black/AA children were four times as likely to die from RCC in the urban area and in the metropolitan area, they were almost three times as likely to die. After controlling for age, sex, and insurance, as the main pRCC prognostic factors, the risk of dying increased in both urban and metropolitan areas. Compared to white children in urban areas, black/AA children were almost nine times as likely to die, adjusted hazard ratio (aHR) = 8.87; 99% CI, 2.77–28.81; *p* < 0.001. In the metropolitan area, there was an increased risk of dying among black/AA children with RCC, compared to their white counterparts, after controlling for age, sex, education, and insurance, aHR = 3.37, 99% CI = 1.35–8.44, *p* = 0.001. Figure 3 illustrates the survival experience of children with RCC in metropolitan versus rural–urban areas.

## 4. Discussion

These data reflect more than four decades of the cumulative incidence of pRCC in the USA. With pRCC being rare, although characterized with poorer survival, we assessed the survival of children diagnosed with this condition. The Cox proportional hazard model was utilized to examine the predictors of survival with specific focus on area of residence as a potential explanatory variable for the racial survival variance in pRCC. To determine the non-confoundability in the model, we applied a multivariable Cox proportional hazard model after testing for the assumption that the hazard ratio remained constant over time.

There are a few relevant findings based on this model. First, males relative to females illustrated a survival disadvantage from pRCC. Secondly, black/AA children relative to white children with RCC had a survival disadvantage. Thirdly, compared to the metropolitan area, children diagnosed with RCC residing in the urban area had a survival disadvantage. Fourthly, area of residence, namely urban and metropolitan, explained in part the survival disadvantage of black/AA children with RCC, indicative of an effect measure modifier, as racial heterogeneity in pRCC racial survival differentials.

Pediatric malignant neoplasms continue to increase, despite improvement in survival in most malignancies, namely ALL, lymphoma, retinoblastoma, thyroid cancer, etc. Survival disadvantages in other malignancies have been observed in AML/CML, brain, and central nervous system tumors (glioblastoma, ependymoma, astrocytoma), as well as renal malignancy, including but not limited to Wilms Tumor and RCC [22,23]. Our demonstration in this sample, which is a large representative sample of pRCC, with a rare malignant neoplasm, is relative to ALL, AML, brain/CNS tumors, lymphoma, and retinoblastoma. However, despite the rare cumulative incidence of pRCC, its survival has been illustrated to be poor, as implicated in these data.

Female children with RCC, compared to their male counterparts, were observed with less mortality, implying a survival disadvantage of male children. These findings of the survival advantage of females with pRCC have been observed in other malignancies, namely leukemia, brain and CNS tumors [22], and lymphoma [23]. Available data attempting to explain the survival disadvantage of males with a malignant neoplasm often fail to provide a reliable and meaningful explanation, including though not limited to social determinants of health (SDHs), Epigenomic Determinants of Health (EDHs) [24], and tumor prognostic factors such as stage, grade, histology, etc., given the sex variance. Holmes et al. observed sex variability in leukemia survival with male children illustrating a survival disadvantage [25]. In this study, hormonal differences explained the survival disadvantage in males, by implicating estrogen as a protective factor against the proliferative pathways in ALL development [25]. The provided explanation was supported by the consistent observation of an increased incidence as well as survival disadvantage in the age groups 10–14 and 15–19. In addition, the survival disadvantage of male children may be driven in part by dietary patterns, unavailable in the SEER Database. However, we were unable to assess dietary habits due to these data source limitations. The implication of diet in the tumor prolific pathway, as well as in the prognosis, is due mainly to a highly methylated diet, which results in DNA methylation involving epigenomic modulations, thereby inhibiting drug response by altering the transcription factors along with impaired protein synthesis, and hence marginalized drug receptors [24].

Epigenomic modulations begin very early in life, commencing at gametogenesis, in utero and post-natal, and reflect everyday circumstances that result in the gene and environment interaction [24]. Specifically, the gene and environment interaction that involves aberrant DNA methylation (mDNA) at the cytosine–phosphate–guanine (CpG) enhancer region of the gene influences the transcription factors and protein synthesis, resulting in abnormal cellular proliferation, implying leukomogenesis [24]. In addition, the histone protein modification by the acetylation process may result in a mutation that reflects an abnormal protein synthesis, either structural, or regulatory, which are implicated in leukomogenesis by restricting DNA access and the subsequent transcriptome impairment [24]. The understanding of epigenomic modulations and the mechanistic process in gene and environment interaction in leukemic genes may facilitate specific risk characterization and induction therapy with demethylase prior to primary therapies in the treatment of pRCC, thus narrowing the black–white risk differences in pRCC mortality [24].

This study has illustrated the survival disadvantage of black/AA children diagnosed with RCC. While the cumulative incidence of RCC is more common among white children, the disproportionate burden of this incidence adversely affects black/AA children. These pRRC data significantly illustrate the burden of survival among black/AA children, indicative of a survival disadvantage, despite controlling for tumor prognostic factors. Holmes et al. observed leukemia, thyroid malignancy, second primary pediatric cancer, and brain and CNS malignancies, with survival consistently observing the survival disadvantage of black/African American children [22,26,27,28,29,30,31,32].

The observed excess mortality of black children with RCC may be explained by SDHs, such as health insurance coverage, early diagnosis, parental education, median income, and disadvantaged neighborhood environmental factors. The previous literature on adult and pediatric malignancies has implicated race as a predisposing factor in cancer survival [23,33,34,35,36,37]. The current study supports previous data on childhood cancer, which associated race as a single and the most predisposing factor to mortality [22,23,33,38,39,40,41,42,43]. Further, the survival disadvantage of black/AA children with RCC based on these data is explained in part by health insurance coverage, which reflects the access and utilization of pediatric Oncology Care. In addition, SDHs, characterized by social inequity, which is the systemic and unfair distribution of social, economic, and environmental conditions related to health, remain as prognostic factors in the pRCC survival disadvantage. In this sample, substantial SDH variables were unavailable in controlling for these potential confounders for racial variance in pRCC survival explanation.

We have clearly demonstrated the exposure effect of area of residence in pRCC survival. Using a dichotomous classification of area of residence, namely urban and metropolitan, we observed a survival disadvantage of children residing in urban areas and diagnosed with RCC. We are unaware of any other findings that implicated area of residence as an environmental factor of RCC prognoses and survival. In this sample, poorer survival was observed in the urban area, which may be attributed to a lack of cancer treatment centers, reduced availability of diagnostic tools, unqualified cancer care providers, and marginalized cancer care providers. In addition, it is plausible that the observed disadvantage of pRCC survival among children in the urban area may be explained by gene–environment interaction, resulting in epigenomic modification, as well as aberrant epigenomic modulations in impaired tumor suppressor genes, p27, p53, and apoptosis cell factors [24].

These data clearly observed area of residence as an effect measure modifier in the relationship between pRCC survival and race of the patients. An effect measure modifier reflects the changes in the crude, unadjusted model relative to the adjusted model. In this representative pRCC data, the hazard associated with survival was augmented, given the interaction between race and area of residence as urbanity. We are unaware of previous literature on the conjoined effect of race and area of residence in pRCC survival. However, these data explain the persistently observed survival disadvantage of black/AA pRCC patients. The observed interaction and its implication in survival may be explained in part by a disproportionate burden of pRCC prognostic factors in urban versus metropolitan areas. The conjoined effect of area of residence and race as an exposure function of pRCC survival illustrated excess risk of dying in the urban area, relative to the metropolitan area.

Despite the novelty of this study, mainly the sample size, reliable statistical modeling, and surrogate epigenomic modulation implication, there are some limitations. First, this study involved a retrospective cohort design of pre-existing data from the NCI registry SEER, which is subject to information, misclassification, and selection biases. However, there is no anticipation of any of these biases as differentials, with respect to exposure and the outcome variables. Secondly, due to the nature of these data, a significant causal correlation cannot be established, implying limited causal inference on the implication of a conjoined effect of area of residence and race in pRCC survival in these findings. While studies have examined disparities in sex, large confidence intervals and insignificant *p* values marginalize inferential applications. Nevertheless, the causal association between race and survival remains, since race predicts pRCC diagnoses, prognoses, and survival. However, the specific therapeutics were not included in these data, but observed in other studies [43,44]. Thirdly, these findings may be driven in part by unmeasured and residual confounding, since no matter how sophisticated statistical software used is, residual confounding remains [22]. Nevertheless, it is highly unlikely that the observation of area of residence as an effect measure modifier in racial variance associated with pRCC survival is driven solely by this unmeasured confounding. Fourthly, our inference is partly limited with respect to racial variance and survival due to the unavailability of data on treatment modalities, namely chemotherapy and radiation, in the SEER registry. Fifthly, the SEER dataset fails to provide substantial information on SDHs at the individual level, implying caution in the interpretation of SDH variables available in the SEER registries [24,44,45]. Failure to apply caution in the interpretation of these variables will result in an ecologic fallacy. Further, the observed differences in urban versus metropolitan areas with respect to survival may be driven by the small sample size of pRCC diagnosed in the urban area, as reflected in the precision values, namely the confidence interval. In effect, given the limitations of the SEER registry in providing the substantial variables on SDHs, our understanding of the interaction between SDHs and epigenomic modulations in cancer requires prospective studies to be conducted in examining socio-epigenomics in racial and sex differentials in pRCC survival, therefore characterizing specific risk and initiating induction therapy prior to surgery, chemotherapy, or radiation.

## 5. Conclusions

pRCC reflects an increased cumulative incidence, annual percent trends in males, and an age of diagnosis between 10 and 14 years, as well as a survival disadvantage of black/AA children. Substantially, neighborhood environmental factors significantly influenced the racial differences in pRCC mortality and survival. These data are suggestive of the conjoined effect of environment and race in pRCC survival, requiring further assessment of gene (DNA sequence)–environment (air quality, toxic waste, SDHs, water quality, etc.) interaction (aberrant epigenomic modulations) in the incidence, therapeutics, prognosis, survival, and mortality in pRCC outcomes.

### 5.1. Key Findings

pRCC indicates an increased cumulative incidence, trends in males, and an age of diagnosis between 10 and 14, as well as survival disadvantages of black/AA children. Additionally, neighborhood environmental factors significantly influenced racial differentials in mortality and survival.

### 5.2. Future Directions

These data are suggestive of the conjoined effect of environment and race in pRCC survival and require further assessment of gene–environment interaction (epigenomic modulation) in incidence, therapeutics, and mortality.

### 5.3. Clinical Implications

At a national level, research observed academic and urban hospitals with higher use of partial nephrectomy as a treatment for RCC than at rural hospitals, which may lead to rural–urban disparities in mortality and exacerbate existing racial disparities in pRCC. There is a suggestion for induction therapy prior to the pRCC standard of care.

## Figures and Tables

**Figure 1 cancers-16-03975-f001:**
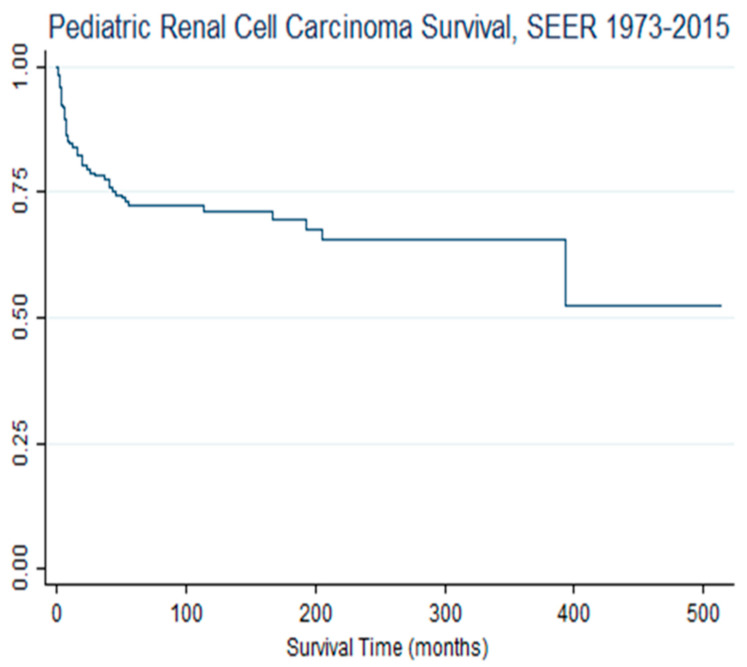
Kaplan–Meier survival proportion of children with renal cell carcinoma, SEER, 1973–2015. Notes and Abbreviations: SEER = Surveillance Epidemiology and End Result. The overall pRCC sample survival is higher than median survival in this sample.

**Figure 2 cancers-16-03975-f002:**
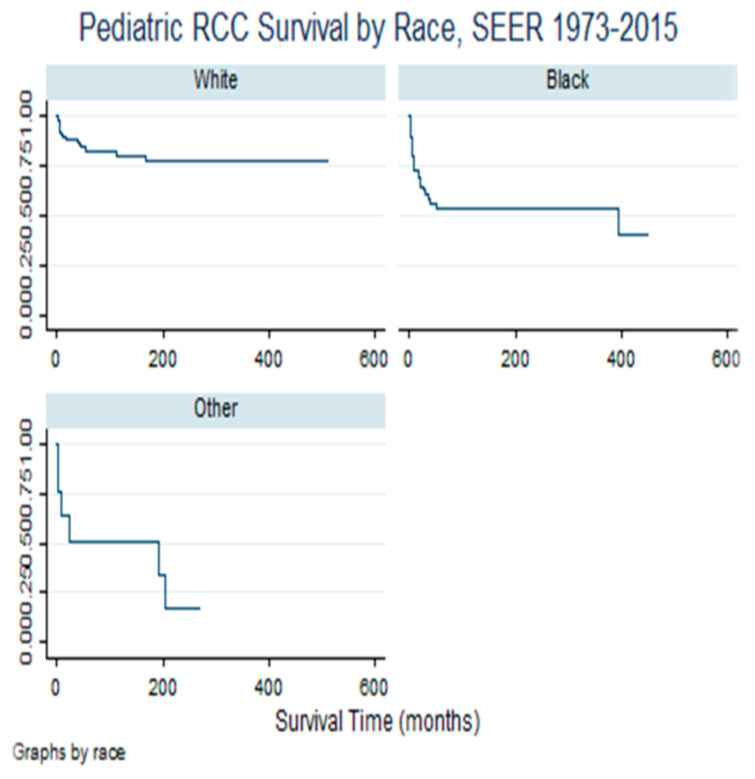
Pediatric Renal Cell Cancer Survival variability by race, Kaplan-Meier Survival Proportion. Notes and Abbreviations: SEER= Surveillance Epidemiology and End Results; RCC= Real Cell Carcinoma. The lower the Kaplan-Meier curve, the more the survival disadvantage, hence excess mortality.

**Figure 3 cancers-16-03975-f003:**
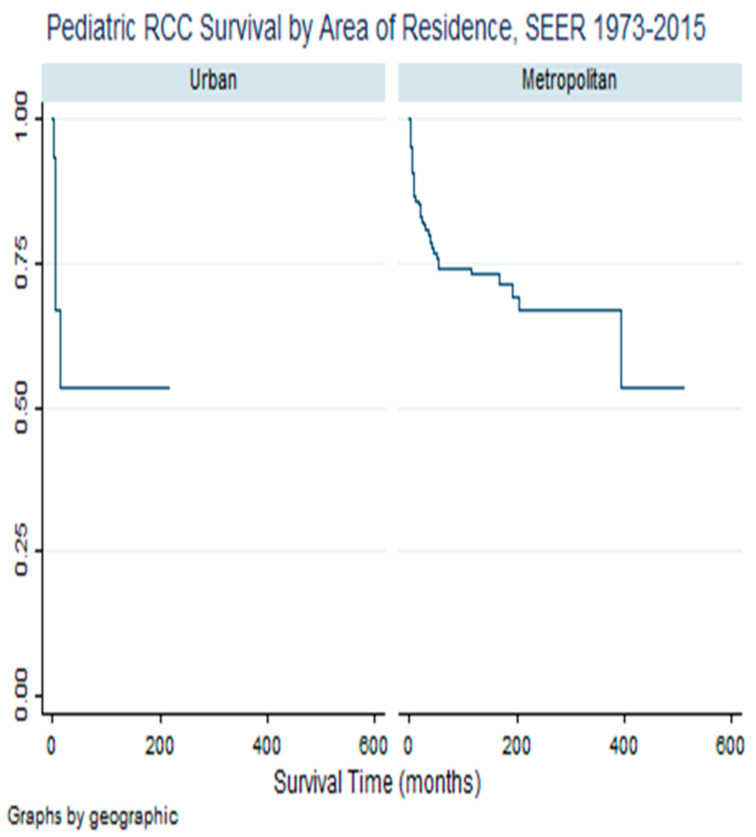
Kaplan-Meier Survival Differential—Pediatric Renal Cell Carcinoma Survival by Urbanity. SEER, 1973–2015. Notes and Abbreviations: RCC = Renal cell carcinoma; SEER = Surveillance and Epidemiology End Result. The survival line in urban area indicates survival disadvantage in this urbanity.

**Table 1 cancers-16-03975-t001:** Study characteristics of pediatric patients (0–19 years) with RCC, stratified by area of residence: Surveillance, Epidemiology, and End Results, 1973–2015.

Variables	Area of Residence (Urbanity)	
	Urban 15 (8.62)	Metropolitan 159 (91.38)
	*n* (%)	*n* (%)
** *Mortality* **		
Alive	8 (53.33)	117 (73.58)
Dead	7 (46.67)	42 (26.42)
** *Race* **		
White	11 (73.33)	105 (66.04)
Black	3 (20.00)	46 (28.93)
Other	1 (6.67)	8 (5.03)
** *Sex* **		
Male	6 (40.00)	83 (52.20)
Female	9 (60.00)	76 (47.80)
** *Age* **		
0	0 (0)	1 (0.63)
01–04	1 (6.67)	8 (5.03)
05–09	0 (0)	20 (12.58)
10–14	5 (33.33)	43 (27.04)
15–19	9 (60.00)	87 (54.72)
** *Tumor Grade* **		
1	0 (0)	8 (5.03)
2	2 (13.33)	27 (16.98)
3	2 (13.33)	17 (10.69)
4	2 (13.33)	9 (5.66)
Unknown	9 (60.00)	98 (61.64)
** *Education* **		
High	2 (13.33)	43 (27.04)
Moderate	3 (20.00)	39 (24.53)
Low	10 (66.67)	77 (48.43)
** *Insurance* **		
Medicaid	2 (13.33)	21 (13.21)
Blank	7 (46.67)	93 (58.49)
Unknown/Uninsured	1 (6.67)	8 (5.03)
Private	5 (33.33)	37 (23.27)
** *Income* **		
Low	5 (33.33)	39 (24.53)
Moderate	0 (0)	45 (28.30)
High	10 (66.67)	75 (47.17)

Abbreviation and notes: *n*, sample size. The cells with “0 (0)” reflect no data available for pRRC patients. The age 0 indicates infants, 0–12 months. The income is indicative of household median income rather than individual income of parents and caregivers.

**Table 2 cancers-16-03975-t002:** Pediatric RCC Survival Risk and Prognostic Factors: Surveillance, Epidemiology, and End Results, 1973–2015.

Variable	HR	95% CI	*p*
** *Area of Residence* **			
Urban	1.00	reference	reference
Metropolitan	0.47	0.21–1.05	0.065
** *Race* **			
White	1.00	reference	reference
Black	2.90	1.58–5.31	0.001
Other	---	---	---
** *Sex* **			
Female	1.00	reference	reference
Male	1.21	0.69–2.11	0.511
** *Age* **			
15–19	1.00	reference	reference
01–04	0.28	0.04–2.05	0.210
05–09	0.50	0.18–1.44	0.200
10–14	0.89	0.47–1.68	0.720
0	---	---	---
** *Tumor Grade* **			
1	0.38	0.11–1.31	0.126
2	0.06	0.01–0.29	<0.001
3	0.69	0.27–1.79	0.446
4	1.00	reference	reference
** *Education* **			
High	1.00	reference	reference
Moderate	0.84	0.36–1.99	0.685
Low	1.51	0.76–2.99	0.235
** *Insurance* **			
Medicaid	1.00	reference	reference
Unknown/Uninsured	2.76	0.39–19.6	0.311
Private	3.79	0.85–16.9	0.081
** *Income* **			
Low	1.00	reference	reference
Moderate	1.03	0.46–2.29	0.946
High	1.11	0.56–2.24	0.751

Abbreviations and Notes: CI, Confidence Interval; HR, Hazard Ratio; type I error tolerance set at 5% (95% CI). A blank reflects no data available for pRRC patients. The age 0 indicates infants, 0–12 months. The income is indicative of household median income rather than individual income of parents and caregivers.

**Table 3 cancers-16-03975-t003:** Conjoined effect of area of residence and race in pediatric RCC survival: Surveillance, Epidemiology, and End Results, 1973–2015.

Models and Variables	Area of Residence					
	Urban			Metropolitan		
	a-HR	99% CI	*p*	a-HR	99% CI	*p*
*Model I (95%CI, p < 0.05)*						
Race (white, black/AA)	4.18	0.84–20.8	0.08	2.78	1.45–5.34	0.002
*Model II*						
Race, age, sex	7.67	0.65–90.9	0.03	3.02	1.28–7.14	0.001
*Model III*						
IIIa. Race, age, sex, education	13.01	0.44–38.81	0.05	3.08	1.28–7.40	0.001
IIIb. Race, age, sex, income	---	---	---	2.92	1.20–7.05	0.002
*Model IV*						
IVa. Race, age, sex, insurance	8.87	2.77–28.10	<0.001	3.24	1.34–7.83	0.001
IVb. Race, age, sex, education, insurance	---	---	---	3.37	1.35–8.44	0.001

Abbreviations and Notes: CI, confidence interval; a-HR, adjusted hazard ratio. Type I error tolerance set at 1% (99% CI). The type I error tolerance for Model I was set at 5%, with a comparable 95% CI. White children were used as a reference group; black/AA was used in the race variable. Due to a small sample size in the urban area, parameter values remained non-estimated as (---).

## Data Availability

This study applied the secondary data from the NCI with Data Used Approval to Laurens Holmes, Jr.

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
