# Peer review of "Pediatric Renal Cell Carcinoma (pRCC) Subpopulation Environmental Differentials in Survival Disadvantage of Black/African American Children in the United States: Large-Cohort Evidence"

_cancers, 2024, doi:10.3390/cancers16233975_

Round 1
Reviewer 1 Report
Comments and Suggestions for Authors
The authors analyzed socio-epidemiologic data from 174 children who were diagnosed with and treated for renal cell carcinoma (RCC) in the period 1971-2015 in the USA. They found that the outcome of the disease was significantly less favorable in Afro American children living in cities and with a worse socio-economic status and low health insurance. The results are well presented. The writing style needs a lot of improvement, because the sentences are redundant and it is difficult for the reader to follow the text. That is why it should be significantly shortened to the essentials in all parts, from the summary (200 words!) and introduction to the conclusion.
Comments on the Quality of English LanguageThe writing style needs a lot of improvement, because the sentences are redundant and it is difficult for the reader to follow the text. That is why it should be significantly shortened to the essentials in all parts, from the summary (200 words!) and introduction to the conclusion.
Author Response
Reviewer 1 Comments and Authors’ Response
R1-1. The authors analyzed socio-epidemiologic data from 174 children who were diagnosed with and treated for renal cell carcinoma (RCC) in the period 1973-2015 in the USA. They found that the outcome of the disease was significantly less favorable in Afro American children living in cities and with a worse socio-economic status and low health insurance. The results are well presented.
Authors’ Response (AR): Thanks immensely for this critical appraisal of this pRCC electroscript (manuscript). The findings are quite impressive, implicative of risk determinants and prevention in pRCC as well as survival improvement in black/AA subpopulation.
R1-2. The writing style needs a lot of improvement, because the sentences are redundant and it is difficult for the reader to follow the text. That is why it should be significantly shortened to the essentials in all parts, from the summary (200 words!) and introduction to the conclusion
AR-2. Thanks for this observation. We have addressed this. However, due to these secondary and pre-exiting data, the abstract required more wordings, 300-400, etc.
Reviewer 2 Report
Comments and Suggestions for Authors
Thank you for the opportunity to review this article. The present study aimed to investigate subpopulation variations in pediatric renal cell carcinoma (pRCC) by: characterizing pRCC by demographic and clinical features; examining the temporal trends as age-adjusted incidence, rate ratio, percent change and annual percent change; and assessing event-free survival (EFS) and survival differences by race and area of residence. The design of this study involved a population-based registry and utilized a retrospective cohort design. The authors found an increased trend of pRCC in men and at age of diagnosis between 10-14, as well as survival disadvantage of black children. Also, the role of some environment factors was highlighted as significant factors of differences in mortality rate.
Although the authors stated several limitations of this study, numerous parts are not supported by appropriate references, while some parts are not understandable (English grammar needs to be checked and improved). The study design is not very clear. Unfortunately, the paper in the present form is not acceptable for publication in Cancers.
The main shortcomings in the manuscript (in addition to already above mentioned) are the following: there are several statements which are not supported by the references; p values are missing; overall the references are old (the reference list has to be updated as much as possible). In addition, a more specific discussion of the obtained data is required.
I recommend this article to be significantly improved before it can be considered for publication in Cancers.
Here are some suggestions that authors might find useful:
1) Please state the exact results as well as p value (when you state that some outcome is higher compared to other one, you need to specify is it statistically significant or not showing the exact p value);
2) In the introduction section you stated the epidemiological data from 2017. Is there any recent data?
3) As currently stated in the introduction section, it seems that there is only one study “at a national level” (rows 94-96). Please confirm and be specific in your statements as much as possible;
4) Please include the reference after the following statement: “Despite the available previous clinical and epidemiological data which correlate higher incidences of pediatric cancer among whites and survival disadvantages among blacks.” (rows 97-99);
5) After the above-mentioned sentence (my comment n.4), you stated: “this correlation may not exist in pRCC incidence and survival at a national level.” Please explain better. In this context, the next sentence, “For example, Rialon et al. found no differences in gender or ethnicity between those who 100 underwent surgical therapy and those who did not.8” However, previously you were talking about incidences of pediatric cancer among whites and black, while here you highlight the gender. So, this part needs to be rewritten. In the current form it does not make sense;
6) You stated: “…this study used the 1973-2015 records of the Surveillance Epidemiology and End Results (SEER) registry….”, I am wondering why until 2015 and not later?
7) Several times in the text there is (REF.) (e.g, rows 120, 133…) So, please include them !
8) What do you mean by “follow-up for vital status”?
9) How was “poverty” determined?
10) You stated the following: “we presented incidence trends and survival estimates for three racial groups (white, black, and other).” However, I do not see any data regarding “other”;
11) The reference is missing after the following statement: “SEER cases are reported to NCI annually each November and there is a completion rate of approximately 98% for all site-specific malignancy, except for melanoma.”
12) Is there any specific reason why the trends data were divided into two registry periods?
13) Please remove the following sentence: “Sex was recoded as male and female.” And use men and women instead of males and females;
14) Please correct the following sentence: “The current study data involved pre-existing cases of children with RCC (n=XXX).”
15) In the results section you mentioned: “Although imprecise….” (Row 271, 279) What do you mean by this statement? Similarly, please explain the following statement too: “….of the parameter precision value…”;
16) In the table 3 some results seem to be missing… You stating that: “Due to small sample size, we were unable to estimate parameter values.” Please explain it better and specify in the materials and methods the exact number of the patients from the registry who have been taken in consideration;
17) I am wondering why the type of health insurance should influence the outcomes? In the same context, as we you are reporting the data on pRC I am wondering how insurance and income could influence it? I suppose you think about those of the parents? The same question is referring to the education level (you reported that also 1 year old children were taking on the consideration)?
18) It is not clear if the following sentence is your own data or not: “First, the proportional morbidity indicated pRCC to be rare compared to Wilms Tumor as well as acute lymphocytic leukemia.”
19) The reference is missing after the following statement: “Pediatric malignant neoplasm continues to increase, despite improvement in survival in most malignancy, namely ALL”
20) Please check and define all abbreviations;
21) Please add the reference after the following sentence: “Survival disadvantage in other malignancies have been observed in AML, brain and central nervous system tumors, as well as renal malignancy, including but not limited to Wilms Tumor and RCC.”
22) You stated: “We have demonstrated in our sample which is a nationally representative sample of pRCC, that this malignant neoplasm is rare, relative to ALL, AML, brain/CNS, lymphoma, and retinoblastoma.” However, I am not sure you are presenting all this data;
23) Again, the reference is missing…. “However, despite the rare cumulative incidence of pRCC, its survival has been shown to be poor, as….”
24) The following sentence is very long and, again, the reference(s) is(are) missing: “Available data attempting to explain the survival disadvantage of males with malignant neoplasm often fails to provide a reliable and meaningful explanation, including though not limited to social determinants of health and biological aggressivity of the tumor given sex variance.”
25) Regrading the following sentence: “Holmes et al. in their publication on leukemia observed sex variable in the survival with male children illustrating survival disadvantage”, please provide the reference and explain better;
26) You stated: “In this study, we utilized hormonal differences to account for the survival disadvantage in males, by implicating estrogen as protective factor against the proliferative pathways in ALL development,” but I am not sure this data is presented…..
27) The following sentence is somewhat confusing: “The provided explanation was supported by the consistent observation of increased incidence as well as survival disadvantage in the age groups 10-14 and 15-19.” Please explain what is “the provided explanation”;
28) Here it is not clear if this is your own finding or some reference is missing: “In addition, the survival disadvantage of male children may be driven in part by dietary patterns(REF).”
29) You stated: “We are unable to assess dietary habits due to data limitations.” Please explain better. There is no data at all or not enough or what…
30) The following statement is not clear, the reference(s) is(are) missing, it does not make sense here…. “The implication of diet in the tumor prolific pathway, as well as in the prognosis, is due mainly to highly methylated diet which results in DNA methylation involving epigenomic changes, thereby inhibiting drug response by altering the transcription factors.” Please check and correct accordingly.
31) The following sentence is not clear: “This study has illustrated the survival disadvantage of black children diagnosed with RCC and followed for this condition.” What do you mean by “followed for this condition”?
Similarly, the following one: “While the incidence of RCC is more common among white children, black children continue to bear the burden of survival indicative a survival disadvantage, despite controlling for tumor prognostic factors, namely tumor size and grade, as well as primary therapies (surgery, chemotherapy, radiation),” especially the part “despite controlling for…” What do you mean by “controlling” tumor size and grade as well as thereapies?
32) “The observed excess mortality of black children with RCC may be explained by social determinants of health, including though not limited to insurance, early diagnosis, parental education, income, and disadvantaged neighborhood environmental factors.” I am not sure that everything here makes sense... please explain it better, in more details, and support by appropriate references;
33) Please provide more details regarding the following statement: “Previous literature in adult malignancies have implicated race as a predisposing factor to survival 17.”
34) “The current study supports previous data on childhood cancer which associate race as a single, most predisposing factor to mortality, 15,16.” This statement is also very general, please be more precise;
35) “The survival disadvantage of black children with RCC based on our data is explained in part by insurance, which reflects access and utilization of oncology care.” Please explain better
36) The same for the following sentence: “In addition, social determinants of health, characterized by social inequity which is the systemic and unfair distribution of social, economic, and environmental conditions related to health.” As well as the following one: “…social determinants of health were not available for us to control for these potential confounders in explaining racial variance in pRCC survival.” To control?! However, you monitored insurance… please be precise in your statements as much as possible;
37) You stated: “…poorer survival was observed in the urban area, which may be attributed to lack of cancer treatment centers, reduced availability of diagnostic tools, unqualified cancer care providers, as well as low numbers of cancer care providers.” In which area is this happening? How often? Is this really true???
38) The following statement: “it is plausible that the observed disadvantage of pRCC survival among children in the urban area may be explained by gene-environment interaction, resulting in epigenomic modification or alteration towards impaired tumor suppressor and apoptosis cell factors,” has to be supported by appropriate references;
39) What do does “an effect measure modifier” mean? Similarly, the following sentence is somewhat confusing and again the reference is missing: “An effect measure modifier reflects the changes in the crude, unadjusted model relative to the adjusted model.”
40) Please explain better the following statement: “…..may be explained by an uneven distribution of tumor prognostic factors in urban versus metropolitan areas.”
41) The following sentence is not clear: “Despite the novelty of this study, mainly the sample size and reliable statistical modelling…” Also, please explain what is the novelty of this study;
42) “However, we do not anticipate any of these biases to be differential with respect to exposure and the outcome,” Please explain better, it is unclear;
43) “(Holmes, androgenic therapy, 2008)”, what does this mean? Should this be a reference?
44) “Nevertheless, it is highly unlikely the observation of area of residence as an effect measure modifier in racial variance associated with pRCC survival is driven solely by these unmeasured confounding.” Why? Please support this statement by findings from the literature;
45) “Failure to apply caution in the interpretation of these variables will cause ecologic fallacy.” What??????????
46) Please remove “in summary” from the conclusion;
47) The reference list includes a lot of very old references….
Comments on the Quality of English LanguageSome parts are not understandable. English grammar needs to be checked and improved.
Author Response
Reviewer 2 – Comments and Authors’ Response
Thank you for the opportunity to review this article. The present study aimed to investigate
subpopulation variations in pediatric renal cell carcinoma (pRCC) by: characterizing pRCC
by demographic and clinical features; examining the temporal trends as age-adjusted
incidence, rate ratio, percent change and annual percent change; and assessing event-free
survival (EFS) and survival differences by race and area of residence. The design of this
study involved a population-based registry and utilized a retrospective cohort design. The
authors found an increased trend of pRCC in men and at age of diagnosis between 10-14, as
well as survival disadvantage of black children. Also, the role of some environment factors
was highlighted as significant factors of differences in mortality rate.
Although the authors stated several limitations of this study, numerous parts are not supported
by appropriate references, while some parts are not understandable (English grammar needs
to be checked and improved). The study design is not very clear. Unfortunately, the paper in
the present form is not acceptable for publication in Cancers.
Reviewer Comment
The main shortcomings in the manuscript (in addition to already above mentioned) are the
following: there are several statements which are not supported by the references; p values
are missing; overall the references are old (the reference list has to be updated as much as
possible). In addition, a more specific discussion of the obtained data is required.
I recommend this article to be significantly improved before it can be considered for
publication in Cancers.
Author’s Response: Thanks for the observed p value, not included in this study. To be very evidential, p value is not the measure of evidence, as point estimate but only reflects the size of the study. For example if the sample size in subpopulations is very small such as, n=10, if the point estimate, hazard ratio (HR) = 9.99, and p is > 0.05, this finding remains clinically and biologically meaningful.
Here are some suggestions that authors might find useful:
R1) Please state the exact results as well as p value (when you state that some outcome is
higher compared to other one, you need to specify is it statistically significant or not
showing the exact p value);
A1: Not applicable in clinical, biologic, public health findings as measure of evidenc , but only sample size.
R2) In the introduction section you stated the epidemiological data from 2017. Is there
any recent data?
A2. Thanks, the SEER data today does not include many social determinants of health variables, which is the reason in using this previous data.
R3) As currently stated in the introduction section, it seems that there is only one study
“at a national level” (rows 94-96). Please confirm and be specific in your statements as
much as possible;
A3: Thanks. Addressed.
R4) Please include the reference after the following statement: “Despite the available
previous clinical and epidemiological data which correlate higher incidences of pediatric
cancer among whites and survival disadvantages among blacks.” (rows 97-99);
A4. Thanks ….addressed.
R5) After the above-mentioned sentence (my comment n.4), you stated: “this correlation
may not exist in pRCC incidence and survival at a national level.” Please explain better.
In this context, the next sentence, “For example, Rialon et al. found no differences in
gender or ethnicity between those who 100 underwent surgical therapy and those who
did not.8” However, previously you were talking about incidences of pediatric cancer
among whites and black, while here you highlight the gender. So, this part needs to be
rewritten. In the current form it does not make sense;
A5: Thanks …addressed.
R6) You stated: “…this study used the 1973-2015 records of the Surveillance
Epidemiology and End Results (SEER) registry….”, I am wondering why until 2015 and
not later?
A6: Some of variables utilized in this findings are no longer available in SEER data commencing 2016.
R7) Several times in the text there is (REF.) (e.g, rows 120, 133…) So, please include
them !
A7. Thnaks , addressed.
R8) What do you mean by “follow-up for vital status”?
A8. Thanks for this observation. The vital status is dead or alive. However we have explained this
R9) How was “poverty” determined?
A9: Thanks! The poverty level is explained by median household income, < $40,000 annual income.
R.10) You stated the following: “we presented incidence trends and survival estimates for
three racial groups (white, black, and other).” However, I do not see any data regarding
“other”;
A10. Thanks. We have addressed this.
R11) The reference is missing after the following statement: “SEER cases are reported to
NCI annually each November and there is a completion rate of approximately 98% for
all site-specific malignancy, except for melanoma.”
A11.Thanks. We have addressed this.
R12) Is there any specific reason why the trends data were divided into two registry periods?
A12. Thanks. The trends is dependent on temporal trends evaluation and the implicatin in temporal incidence and trends.
R13) Please remove the following sentence: “Sex was recoded as male and female.” And
use men and women instead of males and females;
A13. Thanks we have addressed this as male/female, rather than men and female, given the age of the subjects with pRRC.
R14) Please correct the following sentence: “The current study data involved pre-existing
cases of children with RCC (n=XXX).”
A14. Thanks…we have addressed this.
R15) In the results section you mentioned: “Although imprecise….” (Row 271, 279) What
do you mean by this statement? Similarly, please explain the following statement too:
“….of the parameter precision value…”;
A15. Thanks! The application of “imprecise” reflects the random error quantification, which is > 0.05, based on inferential findings by targeting the population of children with RCC. The p value is indicative of the representative sample utilized in any study, as large sample size for all participants (pRCC) and subgroups (Whites/Blacks, etc.)
.
R16) In the table 3 some results seem to be missing… You stating that: “Due to small
sample size, we were unable to estimate parameter values.” Please explain it better and
specify in the materials and methods the exact number of the patients from the registry
who have been taken in consideration;
A16. Thanks. We have addressed this in table 3, within this electroscript (manuscript)
R.17) I am wondering why the type of health insurance should influence the outcomes? In
the same context, as we you are reporting the data on pRC I am wondering how insurance
and income could influence it? I suppose you think about those of the parents? The same
question is referring to the education level (you reported that also 1 year old children
were taking on the consideration)?
A17. Thanks for this observation. The parental or custodian insurance coverage and their education level are directly linked with pRCC patients as the social determinants of health. The understanding of these implications based on parental and custodians SDH directly impacts the pRRC outcomes among children.
R18) It is not clear if the following sentence is your own data or not: “First, the
proportional morbidity indicated pRCC to be rare compared to Wilms Tumor as well as
acute lymphocytic leukemia.”
A18. Thanks. This is observed in SEER data. Reference provided.
19) The reference is missing after the following statement: “Pediatric malignant
neoplasm continues to increase, despite improvement in survival in most malignancy,
namely ALL”
A19. Thanks . Reference is provided.
R.20) Please check and define all abbreviations;
A20. We have addressed this.
R.21) Please add the reference after the following sentence: “Survival disadvantage in other
malignancies have been observed in AML, brain and central nervous system tumors, as
well as renal malignancy, including but not limited to Wilms Tumor and RCC.”
A21. Thanks….Reference provided.
R22) You stated: “We have demonstrated in our sample which is a nationally
representative sample of pRCC, that this malignant neoplasm is rare, relative to ALL,
AML, brain/CNS, lymphoma, and retinoblastoma.” However, I am not sure you are
presenting all this data;
A22. Thanks for this observation. We the SEER data represents this pediatric sub tumors classification. Reference provided.
R23) Again, the reference is missing…. “However, despite the rare cumulative incidence
of pRCC, its survival has been shown to be poor, as….”
A23. Thanks we have addressed this within the electroscript.
R24) The following sentence is very long and, again, the reference(s) is(are) missing:
“Available data attempting to explain the survival disadvantage of males with malignant
neoplasm often fails to provide a reliable and meaningful explanation, including though
not limited to social determinants of health and biological aggressivity of the tumor given
sex variance.”
A24. Thanks! We have addressed this by providing the reference.
R25) Regrading the following sentence: “Holmes et al. in their publication on leukemia
observed sex variable in the survival with male children illustrating survival
disadvantage”, please provide the reference and explain better;
A25. Thanks. We have provided the reference - ISRN Oncol
. 2012:2012:439070. doi: 10.5402/2012/439070. Epub 2012 Apr 3.
Sex variability in pediatric leukemia survival: large cohort evidence
Holmes, L Jr , et al.
R26) You stated: “In this study, we utilized hormonal differences to account for the survival
disadvantage in males, by implicating estrogen as protective factor against the
proliferative pathways in ALL development,” but I am not sure this data is presented…..
A26. Thanks. We have addressed this.
R.27) The following sentence is somewhat confusing: “The provided explanation was
supported by the consistent observation of increased incidence as well as survival
disadvantage in the age groups 10-14 and 15-19.” Please explain what is “the provided
explanation”;
A27. Thanks . we have addressed this.
R.28) Here it is not clear if this is your own finding or some reference is missing: “In
addition, the survival disadvantage of male children may be driven in part by dietary
patterns (REF).”
A28. Thanks. It is only suggestive based on dietary nutrients and cancer incidence.
R29) You stated: “We are unable to assess dietary habits due to data limitations.” Please
explain better. There is no data at all or not enough or what…
A28. Thanks for this comment. We have explained this.
R30) The following statement is not clear, the reference(s) is(are) missing, it does not make
sense here…. “The implication of diet in the tumor prolific pathway, as well as in the
prognosis, is due mainly to highly methylated diet which results in DNA methylation
involving epigenomic changes, thereby inhibiting drug response by altering the
transcription factors.” Please check and correct accordingly.
A30. Thanks .We have simplified this explanation.
R31) The following sentence is not clear: “This study has illustrated the survival
disadvantage of black children diagnosed with RCC and followed for this condition.”
What do you mean by “followed for this condition”?
A31a. Thanks for this comment. We have addressed this, by not using …followed for the disease, since data collection by SEER does not implicate assessing any specific patients with time. This is only instant data collection and incorporation in the cancer registry, SEER.
Similarly, the following one: “While the incidence of RCC is more common among white
children, black children continue to bear the burden of survival indicative a survival
disadvantage, despite controlling for tumor prognostic factors, namely tumor size and
4 grade, as well as primary therapies (surgery, chemotherapy, radiation),” especially the
part “despite controlling for…” What do you mean by “controlling” tumor size and grade
as well as thereapies?
A31b. Thanks for this comment. The cumulative incidence of black/AA pRRC is indicative of the disproportionate burden of pRRC incidence among blacks/AA children. Since the black/AA population size in USA is < 13%, while Whites is > 72%, pRRC in the black/AA population reflects higher burden of incidence within this subpopulation.
The term “controlling for confounding” reflects adjustment by addressing the mortality by Whites and Blacks and adding the risk variables such as household median income and observing again, the initaitl mortality as lower or higher. This is an epidemiologic concepts explained in this publication by Holmes, L Jr: https://www.routledge.com/Applied-Epigenomic-Epidemiology-Essentials-A-Guide-to-Study-Design-and-Conduct/HolmesJr/p/book/9780367556273
R.32) “The observed excess mortality of black children with RCC may be explained by
social determinants of health, including though not limited to insurance, early diagnosis,
parental education, income, and disadvantaged neighborhood environmental factors.” I
am not sure that everything here makes sense... please explain it better, in more details,
and support by appropriate references;
A32. Thanks. Have addressed this in a simplified approach.
R33) Please provide more details regarding the following statement: “Previous literature
in adult malignancies have implicated race as a predisposing factor to survival 17.”
A33. Thanks. We have explained this.
R34) “The current study supports previous data on childhood cancer which associate race
as a single, most predisposing factor to mortality, 15,16.” This statement is also very
general, please be more precise;
A34. Thanks. We have addressed this in the discussion section.
R35) “The survival disadvantage of black children with RCC based on our data is
explained in part by insurance, which reflects access and utilization of oncology care.”
Please explain better
A35. Thanks. We have explained this …!
R36) The same for the following sentence: “In addition, social determinants of health,
characterized by social inequity which is the systemic and unfair distribution of social,
economic, and environmental conditions related to health.” As well as the following one:
“…social determinants of health were not available for us to control for these potential
confounders in explaining racial variance in pRCC survival.” To control?! However, you
monitored insurance… please be precise in your statements as much as possible;
A36. Thanks. We have addressed this.
R37) You stated: “…poorer survival was observed in the urban area, which may be
attributed to lack of cancer treatment centers, reduced availability of diagnostic tools,
unqualified cancer care providers, as well as low numbers of cancer care providers.” In
which area is this happening? How often? Is this really true???
A37. Thanks for this comment. The specific diagnostics in cancer ailments remains challenging in the rural areas today. This observation is ongoing and requires healthcare locations in rural areas in the USA.
A.38) The following statement: “it is plausible that the observed disadvantage of pRCC
survival among children in the urban area may be explained by gene-environment
interaction, resulting in epigenomic modification or alteration towards impaired tumor
suppressor and apoptosis cell factors,” has to be supported by appropriate references;
A38. Thanks . Reference provided: https://www.routledge.com/Applied-Epigenomic-Epidemiology-Essentials-A-Guide-to-Study-Design-and-Conduct/HolmesJr/p/book/9780367556273
R39) What do does “an effect measure modifier” mean? Similarly, the following sentence
is somewhat confusing and again the reference is missing: “An effect measure modifier
reflects the changes in the crude, unadjusted model relative to the adjusted model.”
A.39. Thanks ! The effect measure modifier (EMM) reflects heterogeneity, which is the implication of the 3rd variable on the causal pathway in outcome modification such as pRCC mortality and survival in this study- https://www.routledge.com/Applied-Biostatistical-Principles-and-Concepts-Clinicians-Guide-to-Data-Analysis-and-Interpretation/HolmesJr/p/book/9780367560072
R40) Please explain better the following statement: “…..may be explained by an uneven
distribution of tumor prognostic factors in urban versus metropolitan areas.”
A40. Thanks. Simply explained as pRRC diagnostics, specific treatment and prevention.
R41) The following sentence is not clear: “Despite the novelty of this study, mainly the
sample size and reliable statistical modelling…” Also, please explain what is the novelty
of this study;
A.41. Thanks …explained as the environmental determinants of pRCC mortality and survival as surrogate aberrant epigenomic modulations.
R42) “However, we do not anticipate any of these biases to be differential with respect to
exposure and the outcome,” Please explain better, it is unclear;
5
A42. Thanks. The non-differential bias does not affect subpopulation differentials in the point estimate such as hazard ration or risk ratio.
R43) “(Holmes, androgenic therapy, 2008)”, what does this mean? Should this be a
reference?
A.43.Tthanks . reference included. https://pubmed.ncbi.nlm.nih.gov/17486111/
R44) “Nevertheless, it is highly unlikely the observation of area of residence as an effect
measure modifier in racial variance associated with pRCC survival is driven solely by
these unmeasured confounding.” Why? Please support this statement by findings from
the literature;
A44. Thanks . The unmeasured confounding is challenging and unexplainable .
R45) “Failure to apply caution in the interpretation of these variables will cause ecologic
fallacy.” What??????????
A45. Thanks . ecologic fallacy reflects a group level data such as household income level and education as group variables. https://www.taylorfrancis.com/books/mono/10.4324/9781315369761/applied-epidemiologic-principles-concepts-laurens-holmes-jr
R46) Please remove “in summary” from the conclusion;
A46. Thanks . We have deleted summary and maintains the “conclusion”
R47) The reference list includes a lot of very old references….
A.47: thanks for this observation. We are utilizing current references but retaining the already utilized references in this Electroscript based on the data utilization, 1973-2015.
Reviewer 3 Report
Comments and Suggestions for Authors
I congrat the authors for a well written in focus study. I only have some minor comments:
Line 66: this sentence seems to be contradicting the remaining affirmations in the paper, with a higher incidence of RCC in white children.
Could you please clarify which states are present in the NCI’s SEER registry?
line 168: there is a typo: (xxx)
line 337: there is a typo: ref
Author Response
Reviewer 3 Comments and Authors Response
Reviewer Comments; I congrat the authors for a well written in focus study. I only have some minor comments:
Line 66: this sentence seems to be contradicting the remaining affirmations in the paper, with a higher incidence of RCC in white children.
Could you please clarify which states in the NCI’s SEER registry?
line 168: there is a typo: (xxx) are present
line 337: there is a typo: re
Authors Response (AR): Thanks immensely for this observation and comments. We have addressed the states involved in this pRRC in the materials /method section, as well as addressed the typos.
Reviewer 4 Report
Comments and Suggestions for Authors
The reviewer understands that Holmes Jr. et al. have presented a manuscript entitled "Subpopulations Environmental Differentials in Survival Disadvantage of Black/African American with Pediatric Renal Cell Carcinoma (pRCC) in the United States: Large Cohort Evidence (SEER Data)". The reviewer has a few suggestions and they would like to request authors to kindly answer all the questions by updating their manuscript.
1) Abstract is missing. Please provide a detailed abstract.
2) An introduction is missing. Please provide an in depth introduction along with a detailed literature review. Cite all the relevant papers in the Literature Review section. Provide a table mentioning the relevant cited work from the literature. Add two additional columns to that table. Please mention positive aspects mentioned in those cited works. In the second column, please mention possible limitations of all the cited works. Please do cite references in that table for your reader audience.
3) Please add a detailed paragraph (last paragraph) in the introduction mentioning what is covered and what is the research gap in the literature. Why did you choose this topic and what are your contributions to the scientific world through this manuscript? Please do emphasize on highlights of your work.
4) Please rewrite materials and methods in detail with more in depth description.
5) Please justify (arrange text like a book) your written text to make it look professional.
6) Please do provide a list of factors or parameters affecting your methodology and work. Please mention how you wok on them in the future.
7) Please mention error %.
8) Rename Conclusion as Conclusions and Future Works and kindly rewrite this section.
9) Please number each section as 1, 2, 3, 4, 5, 6...etc.
10) Please do add more 50 to 70 citations. It will help you to get more in depth kmowledge of what is missing in the literature.
11) Do read any MDPI Cancers published manuscripts. Try to understand the writing sequence. For example, Title, Abstract, Keywords, Introduction, Literature Review, Materials and Methods, Results and Discussions, Limitations, Conclusions andFuture Works, References etc. Your writing flow is currently improper. Kindly update the same.
12) All the figures are blurry. Please provide at least 5-6 figures. All the figures should be in the HD format.
Author Response
Reviewer 4 Comments and Authors Response
The reviewer understands that Holmes Jr. et al. have presented a manuscript entitled
"Subpopulations Environmental Differentials in Survival Disadvantage of Black/African
American with Pediatric Renal Cell Carcinoma (pRCC) in the United States: Large Cohort
Evidence (SEER Data)". The reviewer has a few suggestions and they would like to request
authors to kindly answer all the questions by updating their manuscript.
Reviewer (1): Abstract is missing. Please provide a detailed abstract.
Authors Response (AR1). Thanks for this observation. Abstract was submitted, and please review the abstract.
R2) An introduction is missing. Please provide an in depth introduction along with a detailed
literature review. Cite all the relevant papers in the Literature Review section. Provide a
table mentioning the relevant cited work from the literature. Add two additional columns to
that table. Please mention positive aspects mentioned in those cited works. In the second
column, please mention possible limitations of all the cited works. Please do cite references
in that table for your reader audience.
AR2. Thanks for this observation. Introduction is included in this study on pRCC. Please observe the introduction. This introduction indicates the pRCC issues, what is known and what requires to be known, and how this study address what needs to be known.
R3) Please add a detailed paragraph (last paragraph) in the introduction mentioning what is
covered and what is the research gap in the literature. Why did you choose this topic and
what are your contributions to the scientific world through this manuscript? Please do
emphasize on highlights of your work.
AR3. Thanks for this observation. We have addressed this, by incorporating some sentences in the last paragraph of the introduction.
R4) Please rewrite materials and methods in detail with more in depth description.
AR4.Thanks for this observation. The materials and method section of a scientific paper needs to be brief, with substantial explanation of the approach in data acquisition or collection, variables ascertainment, sample size, data description and analysis.
R5) Please justify (arrange text like a book) your written text to make it look professional.
AR5. Thanks for this observation. We have addressed this very carefully: Below please observed one’s publication:
R6) Please do provide a list of factors or parameters affecting your methodology and work.
Please mention how you wok on them in the future.
AR6. Thanks for this observation. This is a clinical and public health-based findings , and not methodologic appraisal. This study describes the limitations associated with these findings.
R7) Please mention error %.
AR7. Thanks for this observation. This implication in clinical and public health research and findings in inapplicable.
R8) Rename Conclusion as Conclusions and Future Works and kindly rewrite this section.
AR8. Thanks for this observation. The conclusion of any clinical finding is dependent on inference. We have deleted the “summary” based on a the conclusion of the findings.
R9) Please number each section as 1, 2, 3, 4, 5, 6...etc
AR9. The clinical and population-based studies and findings is relevant to such listings, but reflects, introduction, materials/methods, results, discussion and conclusion.
R10) Please do add more 50 to 70 citations. It will help you to get more in depth knowledge
of what is missing in the literature.
AR10. thanks for this observation. We have increased the publications in support of these findings to 35 citations, due to limited publications on pRRC.
R11) Do read any MDPI Cancers published manuscripts. Try to understand the writing
sequence. For example, Title, Abstract, Keywords, Introduction, Literature Review,
Materials and Methods, Results and Discussions, Limitations, Conclusions andFuture
Works, References etc. Your writing flow is currently improper. Kindly update the same.
AR11.Thnaks for this observation. Scientific publication, which reflects MDPI is relevant to this observation of the reviewers, which is implied in this pRCCs electroscript as manuscript. However, “future works’ is not applicable in this pRCCs.
R12) All the figures are blurry. Please provide at least 5-6 figures. All the figures should be
in the HD format
AR12. Thanks for this observation. The application of STATA graphics is utilized in this electroscript. However the two figure are very relevant in this publication while tables are visualized in the observation of these findings.
Round 2
Reviewer 2 Report
Comments and Suggestions for Authors
I think that the authors have adequately addressed the comments in the revised version of the manuscript, although the answers are very short and very often superficial without detailed indication on the changes that have been made in the text. I have no further comments.
Comments on the Quality of English LanguageNo changes compared to the previous version of manuscript
Reviewer 4 Report
Comments and Suggestions for Authors
I accept the updated version of the manuscript.